# Minimum acceptable dietary intake among children aged 6–23 months in Ethiopia: A systematic review and meta-analysis

**Gizachew Ambaw Kassie** [ID]*[☯], **Amanuel Yosef Gebrekidan**[☯], **Eskinder Yilma Enaro**[☯],
**Yordanos Sisay Asgedom** [ID][☯]

School of Public Health, College of Health Science and Medicine, Wolaita Sodo University, Wolaita Sodo,
Ethiopia

☯ These authors contributed equally to this work.
* gizachewambawkase@gmail.com

## Abstract

### Background

In the absence of minimum acceptable diet, children aged 6–23 months are vulnerable to malnutrition. Not feeding at least the minimum acceptable diet is a major global problem, particularly in developing countries. Even though many studies have been conducted in Ethiopia there are inconsistencies. Therefore, this review aimed to estimate the pooled prevalence of a minimum acceptable diet in Ethiopia.

### Methods

Published articles from various electronic databases, such as PubMed/MEDLINE, EMBASE, Google Scholar, and Science Direct were systematically searched. All cross-sectional studies conducted on the minimum acceptable diet of children aged 6–24 months and published up to October 30/2021 were included in this review. Data were extracted using an Excel spreadsheet and analyzed using STATA version 14.1. The random-effects model was used to estimate the pooled prevalence, and a subgroup analysis was performed to identify the possible source of heterogeneity. Begg's and Egger's tests were used to identify possible publication bias.

### Results

Nine cross-sectional studies involving 4,223 participants were included. Significant heterogeneity was observed across the studies ($I^2 = 99.4\%$). The pooled prevalence of minimum acceptable diet in Ethiopia was found to be 25.69% (95% CI: 11.96, 39.41)

### Conclusion

This review revealed that the minimum acceptable dietary intake among children aged 6–23 months in Ethiopia was relatively low; only 1 in 4 of children met the minimum acceptable diet. This indicates that the government should promote child feeding practices according to guidelines to increase the proportion of children with a minimum acceptable diet.

**Data Availability Statement:** All relevant data are within the manuscript and its Supporting Information files.

**Funding:** The author(s) received no specific funding for this work.

**Competing interests:** The authors have declared that no competing interests exist.

**Abbreviations:** CI, Confidence Interval; EDHS, Ethiopia Demographic Health Survey; JBI, Joanna Briggs Institute; MAD, Minimum Acceptable Diet; SDGs, Sustainable Development Goals; PRISMA, Preferred Reporting Items for Systematic Reviews and Meta-Analyses; SNNPR, South Nations Nationalities and Peoples regional of Ethiopia; WHO, Word Health Organization.

## Introduction

Proper feeding practice in children aged 6–24 months significantly reduces the morbidity and mortality caused by malnutrition [1, 2]. The age of 6–24 months is a critical window for improving child nutrition because deficits acquired by this age are difficult to reverse [3]. Therefore, providing a minimum acceptable diet including the recommended diversification and frequency of foods according to age is cost-effective strategy for proper growth and development of children [4]. Appropriate supplementation of breastfeeding with nutritious commentary foods can reduce stunting among children by approximately 20% [5].

The World Health Organization (WHO) recommends that appropriate complementary feeding should start with semi-solid, or soft foods, minimum meal frequency, minimum dietary diversity, minimum acceptable diet, and consumption of iron-rich or iron-fortified foods at the age of six months [6]. The minimum acceptable diet is a composite indicator comprising minimum dietary diversity and minimum meal frequency indicators. It is an indicator used to assess infant and young child feeding practices during childhood [7].

The minimum acceptable diet feeding practices for infants and children is not met in many developing countries including Ethiopia. For example, in Africa less than one-third and one-half of the children aged between 6 and 23 months met the minimum criteria for dietary diversity and meal frequency, respectively. Therefore, the prevalence of stunting in many developing country including African countries is still increasing by threefold during the first two years of life [1]. In Ethiopia, the minimum acceptable diet has increased slightly from 7% to 11% from the 2016 EDHS to 2019 EMDHS reports. The proportion minimum acceptable dietary intake among children is highest in Addis Ababa (28%) and lowest in Somali (1%) followed by Afar (4%) and Amhara (6%) [8].

The global burden of stunting, wasting and overweight among children under five years of age was 149 million, 45 million and 38.9 million by the year 2020, respectively [2]. In Ethiopia, 37% of children under the age of five are stunted, of which 12% are severely stunted, 7% are wasted, and approximately 2% are overweight [8]. Most of the burden of childhood undernutrition can be explained by the absence of proper infant and young child feeding practices during the first two years. In Ethiopia 50–70% of deaths in children under-five years of age are attributable to inappropriate complementary feeding practices. This indicates that feeding of the minimum acceptable diet is a major problem in Ethiopia [9, 13].

Healthy nutrition is one of the vital preconditions for the achievement of sustainable development goals; because the first three icons of seventeen Sustainable Development Goals (SDGs) are directly related to nutrition [10]. To end child under nutrition by 2030 the government of Ethiopian has been tried to improve child feeding practices by implementing the national nutrition program of child feeding practices and a multi-sectoral plan of nutrition intervention [11]. In addition, the recently endorsed 2019 Food and Nutrition Policy is one of the strategy which is aimed to achieve optimal nutritional status throughout the life cycle via coordinated implementation of nutrition-specific and nutrition sensitive interventions were the major activities under implementation [12].

Although, many studies were conducted on minimum acceptable diet to determine the magnitude and determining factors in Ethiopia. However, there were discrepancy and inconsistent across those studies and offered district level data. The results from those studies indicated that children aged 6–23 months which meets minimum acceptable diet in Ethiopia ranges from 8.4%-74% [13, 14]. As a result of that variability across studies pooled prevalence of minimum acceptable diet is needed. Moreover, inclusive estimates of the extent of minimum acceptable diet are needed for the programmatic intervention of malnutrition within the

context of ensuring minimum acceptable diet. Therefore, the aim of this meta-analysis was to estimate the pooled prevalence of minimum acceptable diet in Ethiopia.

The evidences from this study will give the update pooled estimate of minimum acceptable diet to policy makers and health planners of Ethiopian government relevant bodies. Those concerned bodies as input to design appropriate interventions to improve minimum acceptable diet (minimum dietary diversity and minimum meal frequency) of children aged 6–23 months.

## Methods

### Searching strategy and study identification

A systematic review and meta-analysis was designed to estimate the pooled prevalence of a minimum acceptable diet among children aged 6–23 months in Ethiopia. All published research reports on the minimum acceptable diet among children aged 6–23 months in Ethiopia until October 2021 were searched. We systematically and thoroughly explored to identify published and unpublished research findings related to prevalence and associated factors of minimum acceptable diet among children aged 6–23 months in Ethiopia from all electronic databases, including Medline, Google Scholar, HINARI, AJOL, direct Google search for gray literature and Cochrane library. Our searches were restricted by age (children 6–23 months age) and by country (studies conducted only in Ethiopia). The search was carried out using the following key words "prevalence", "minimum acceptable diet among children", OR "meal frequency" OR "dietary diversity score" AND "Ethiopia". All published articles until October 30, 2021 were included in the systematic review and meta-analysis. We followed Preferred Reporting Items for Systematic Reviews and Meta-Analyses (PRISMA) guideline during the systematic review [15]. The protocol of this systematic review and meta-analysis was not registered

### Inclusion criteria

**Population.**   Study conducted on children aged 6–23 months

**Study setting.**   Only studies conducted in Ethiopia were included in this review.

**Publication.**   All articles published both in peer-reviewed journals and grey literature.

**Study design and language.**   All original cross-sectional articles that report the prevalence of minimum acceptable diet among children aged 6–23 months published by English language in Ethiopia were included.

**Exclusion criteria.**   Studies unable to access full text, not in the English language, and studies which did not report specific primary outcomes of interest were excluded from this study.

### Outcomes measurement

The outcome of the current study was the prevalence of children with a minimum acceptable diet which is a composite indicator of minimum dietary diversity and minimum meal frequency. The proportion of children aged 6–24 months who received a minimum diversified diet and minimum meal frequency (apart from breast milk) [16].

### Data abstraction

All studies retrieved from all databases were imported into Endnote Version X7 and duplicate articles were manually removed. Data were extracted by two independent reviewers (GA and AY) using Excel spreadsheet software. Initially title and abstract of the papers were screened for eligibility criteria. After screening of the abstract and titles full text of papers were also

screened for eligibility criteria. The data extraction format includes primary author, publication year, region where the study was conducted, sample size, total number of cases, prevalence of minimum acceptable diet, response rate, and quality rating.

## Quality assessment of the studies

The Joanna Briggs Institute (JBI) tool adapted for cross-sectional studies quality assessment was used to assess the quality of each study [17]. The tool has the following items as criteria for appraisal: includes; appropriateness of the sample frame to address the source population, appropriate recruiting of study participants, sample size sufficiency, description of study subjects and setting, data analysis with sufficient coverage of sample, valid method for identification of the condition, measuring the condition using a standard reliable and consistent method for all participants, use of appropriate statistical analysis, and response rate adequacy or appropriate handling of low response rate. Two independent reviewers (GA and AY) critically evaluated each study. Disagreements between the reviewers were resolved through discussion. If they did not agree, a third reviewer (YS) was engaged to resolve discrepancies between the independent reviewers. Then after, studies with a total score of ≥50% of the quality assessment checklist criteria were considered as low risk and were included in the final review and analysis.

## Statistical analysis

After extraction, the data were entered into a computer using an Excel spreadsheet and then imported to STATA 14 software for further analysis. Heterogeneity across the studies reporting prevalence was checked using the inverse variance ($I^2$) statistical test. $I^2$ with Cochran Q statistic values of 0%, 25%, 50%, and 75% were assumed to represent no, low, medium, and high heterogeneity, respectively at p-value of <0.05. Since, significant heterogeneity was detected between studies (p<0.01, $I^2$ >99.4%), random effects meta-analysis model was used to estimate the pooled prevalence of minimum acceptable diet. To visualize the presence of heterogeneity subjectively forest plot was used. To examine potential discrepancies across the studies included in the analysis and to identify sources of variation subgroup analyses and meta-regression were conducted. The findings of the study are presented using forest plots with corresponding prevalence and 95% confidence intervals. Evidence of publication bias was also assessed using both Egger's and Begg's tests with p-value of less than 0.05 as a cut-off point.

## Results

### Selection of studies

A total of 325 articles were retrieved from the electronic databases and other sources. Titles and abstracts were screened and duplicated or irrelevant articles were removed using End-Note × 7. Thus, 260 duplicate articles were removed. Of the remaining 65 articles, 35 articles were excluded because their titles and abstracts were not in- line with our inclusion criteria (not reporting the outcome of the interest, studies conducted outside Ethiopia), and 21 articles were excluded for reasons (studies conducted outside Ethiopia, insufficient data, study designs other than cross-sectional, study objective not related and outcome of interest is not reported). Finally, a total of nine articles were included in this systematic review and meta-analysis. The detailed selection procedures were described in (Fig 1).

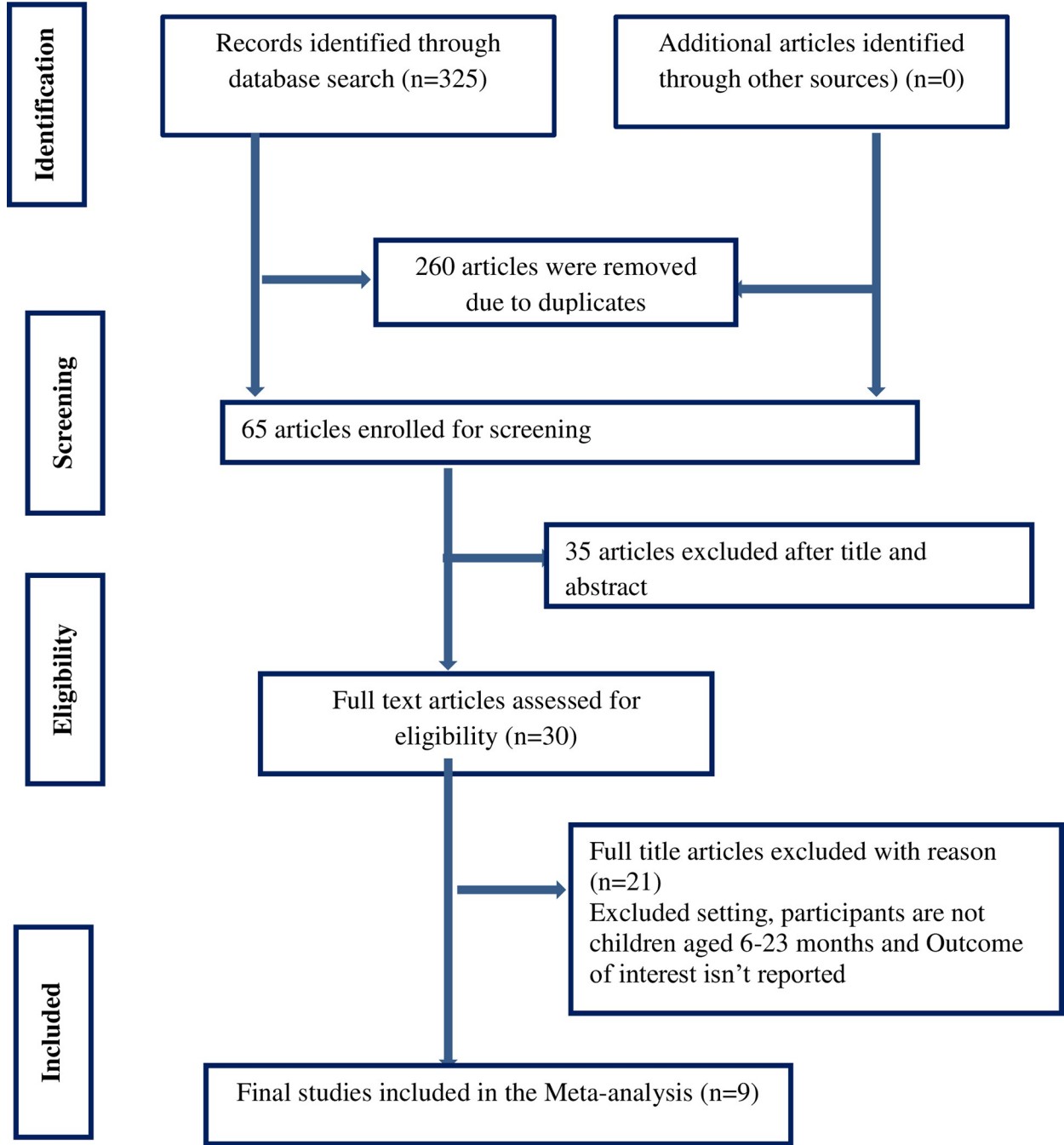

**Fig 1. PRISMA flow diagram of articles screened and the selection process on minimum acceptable diet among children aged 6–23 months in Ethiopia.**

## Characteristics of included studies

Nine cross-sectional studies were included in the final systematic review and meta-analysis. All the studies included in this meta-analysis were published between 2017 and 2021. A total of 4,223 children aged 6–23 months participated in the analysis with a minimum sample size of 200 [18] and a maximum of 662 [19]. Of the nine articles included in this systematic review

**Table 1. Summary of characteristic of nine studies included in the systemic review and meta-analysis on minimum acceptable diet among children aged 6–23 months in Ethiopia, 2022.**

| No | Authors | Publication Year | Region | Study setting | Study design | Sample size | Children who met MAD | Prevalence of MAD (%) |
|---|---|---|---|---|---|---|---|---|
| 1 | Yisak et al. [23] | 2021 | Amhara | Urban | Cross-sectional | 287 | 100 | 34.8 |
| 2 | Birie et al. [20] | 2021 | Amhara | Urban | Cross-sectional | 430 | 54 | 12.6 |
| 3 | Abebe et al. [13] | 2021 | Addis Ababa | Urban | Cross-sectional | 575 | 419 | 74.6 |
| 4 | Mulat et al. [21] | 2019 | Amhara | Rural | Cross-sectional | 505 | 49 | 8.6 |
| 5 | Molla et al. [22] | 2021 | Amhara | Urban | Cross-sectional | 531 | 168 | 31.6 |
| 6 | Gizaw et al. [18] | 2019 | Oromia | Rural | Cross-sectional | 200 | 27 | 13 |
| 7 | Feleke et al. [19] | 2020 | SNNPR | Rural | Cross-sectional | 662 | 235 | 35.5 |
| 8 | Dangura et al. [14] | 2017 | SNNPR | Rural | Cross-sectional | 417 | 35 | 8.4 |
| 9 | Kassa et al. [24] | 2016 | Oromia | Rural | Cross-sectional | 626 | 75 | 12.3 |

and meta-analysis only three regions and one administrative city of Ethiopia was represented. Of the total studies included; four studies were conducted in Amhara regional state [20–23]; two studies in South Nations Nationalities and Peoples of Ethiopia national regional state (SNNPR) [14, 19]; two studies at Oromia national regional state [18, 24] and one study from Addis Ababa city administration [13]. Three studies were conducted in the urban areas and the rest six were conducted at the rural areas (Table 1).

## Pooled prevalence of minimum acceptable diet in Ethiopia

The pooled prevalence of minimum acceptable diet in Ethiopia was 25.69% (95% CI 11.96, 39.41). As shown in the forest plot, statistically significant heterogeneity was observed ($I^2$ = 99.4%; $p < 0.001$). Therefore, we estimated the pooled prevalence using random effects models. In addition, the significant magnitude of heterogeneity also indicates the need to conduct subgroup analysis to identify the sources of heterogeneity across the studies (Fig 2).

## Subgroup analysis

In this meta-analysis subgroup analysis was performed based on the study area (region), study years, and study setting to identify possible sources of heterogeneity (Table 2). Regarding the pooled prevalence by region the lowest pooled prevalence of minimum acceptable diet was reported in Oromia regional state 12.47% (95% CI 10.19, 12.74) and the highest was both in Amhara regional state 21.74% (95% CI 9.63, 33.86) and SNNPR region 21.92% (95% CI 4.64, 48.5) in which the prevalence is almost similar on the two regions (Fig 3). Subgroup analysis by study setting also indicated that the pooled prevalence of the minimum acceptable diet was 47% (17.3. 76.8) among those studies employed in urban areas and 15% in rural areas. There was increment of minimum acceptable diet starting from the studies published before 2020 15.5% (6.74, 24.3) and after 2020 38.4% (9.41, 47.39) (Table 2). In addition to subgroup analysis, publication bias was assessed using both Begg's and Egger's tests. The results of the Begg and Egger tests indicated that there was no identified among studies included to estimate the pooled prevalence of minimum acceptable diet at p–value of (p = 0.055) and (p = 0.066), respectively. The graphical distribution of the funnel plot also shows evidence of symmetry (Fig 4).

## Meta regression

Meta regression was performed by considering continuous variables to identify associated factors with the pooled prevalence of minimum acceptable diet. Sample size and years of

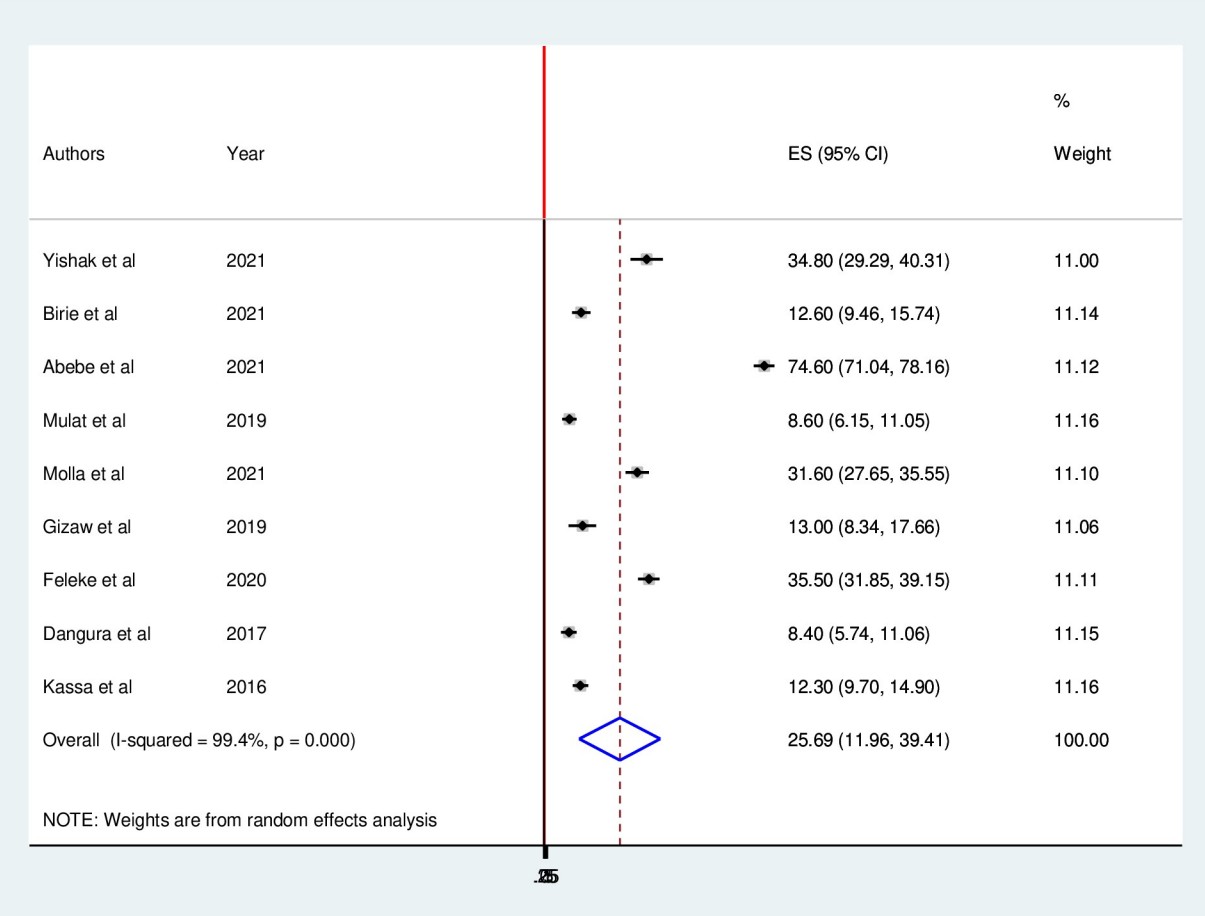

**Fig 2. Forest plot for pooled prevalence minimum acceptable dietary intake among children aged 6 to 23 months in Ethiopia.**

publication were considered in the meta-regression analysis. However, the meta-regression indicated that there was no a significant association between the pooled prevalence of the minimum acceptable diet with years of publication and sample size (Table 3).

**Table 2. Subgroup analysis for the pooled prevalence of minimum acceptable diet among children aged 6–23 months in Ethiopia.** 2022.

| Subgroups | Number of studies | Prevalence(95%CI) | Heterogeneity statistics | p-value | I2 | Tau-squared |
|---|---|---|---|---|---|---|
| **Regions** | | | | | | |
| Amhara | 4 | 21.74 (9.63, 33.86) | 144.12 | 0.0000 | 97.9 | 148.8 |
| Oromia | 2 | 12.47 (10.19, 12.74) | 0.07 | 0.0001 | 0 | 0 |
| SNNPR | 2 | 21.92 (4.64, 48.5) | 138.46 | 0.0001 | 99.4 | 364 |
| **Years of publication** | | | | | | |
| Before 2020 | 5 | 15.5 (6.74, 24.3) | 669.2 | 0.0000 | 99.6 | 870.75 |
| After 2020 | 4 | 38.4(9.41, 47.39) | 170.3 | 0.0001 | 97.7 | 97.71 |
| **Study setting** | | | | | | |
| Urban | 3 | 47.0(17.28. 76.78) | 294.49 | 0.0000 | 99.3 | 684.1 |
| Rural | 6 | 15.0(7.8, 22.22) | 170.46 | 0.0001 | 97.1 | 78.39 |

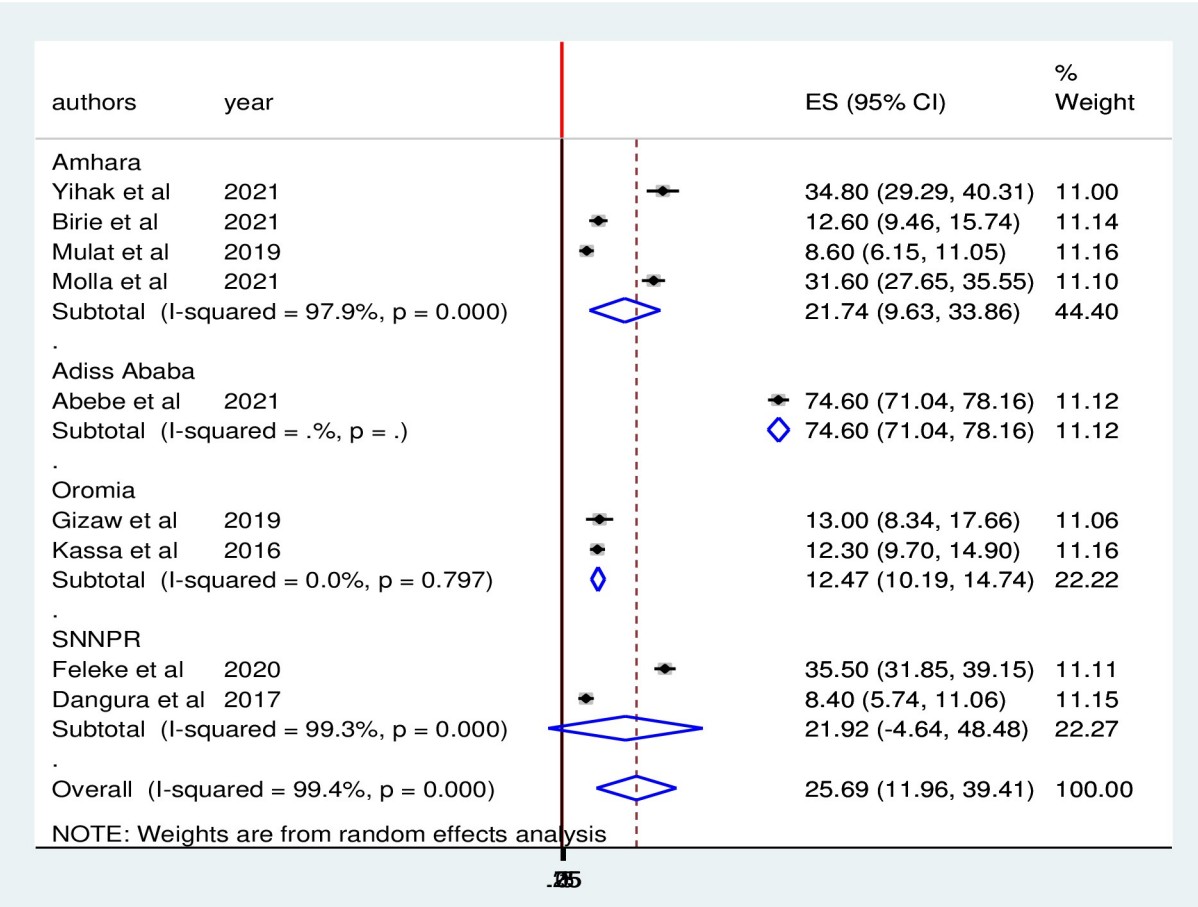

**Fig 3. Subgroup analysis forest plot for pooled prevalence of minimum acceptable dietary intake by regions among children aged 6 to 23 months in Ethiopia.**

## Sensitivity analysis

Sensitivity analysis was done to evaluate the effect of single study on the pooled prevalence of minimum acceptable diet among children aged 6–23 months by excluding each study step-by-step. The results showed that no single studies make a significant difference in the pooled prevalence minimum acceptable dietary intake (Fig 5).

## Discussion

This systematic review and meta-analysis aimed to estimate the pooled prevalence of minimum acceptable diet in Ethiopia. This study revealed that the pooled prevalence of minimum acceptable diet in Ethiopia was found to be 25.69% (95% CI 11.96, 39.41). This Indicates that most children aged 6–23 months in Ethiopia were not taking the minimum acceptable diet according to global recommendations [29]. The result of this meta-analysis was consistent with the national Demographic and Health surveys in China (27.4%) [25], Democratic Republic of Congo (33%) [26], Ghana (14%) [27], and Indonesia based on DHS data (29%) [28].

However, it was higher than according to the 2019 mini Ethiopian demographic and health survey report (11%) [8], a national survey conducted in low- and middle-income countries (10.1%) [29] and multilevel analysis conducted in sub-Sahara Africa (9.98%) [30]. This might

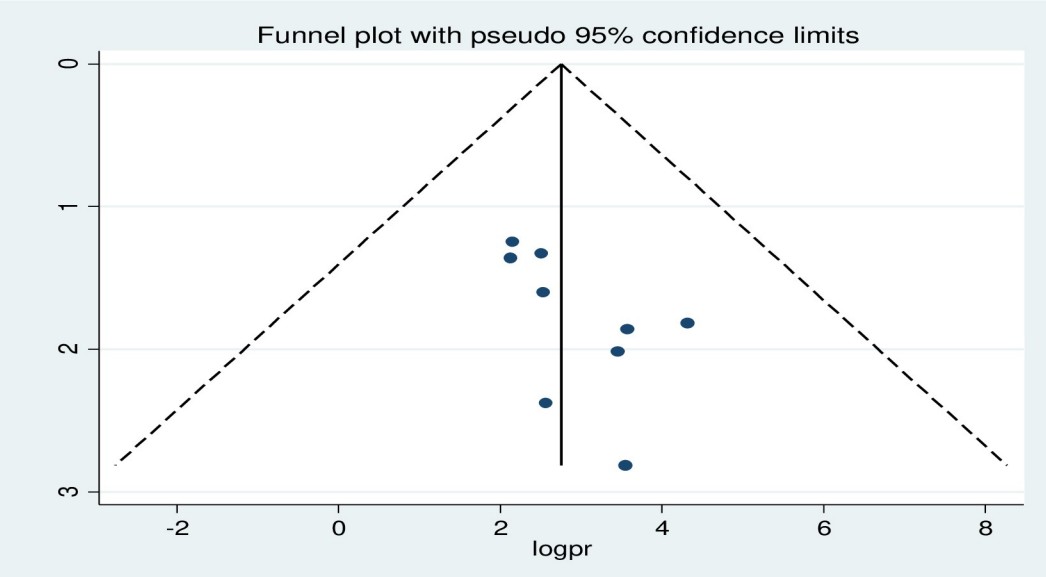

**Fig 4. Funnel plot for publication bias on minimum acceptable dietary intake among children aged 6 to 23 months in Ethiopia.**

be the fact that all the above studies were conducted based on demographic and health survey data of the countries in which they were conducted by including a sampled data from all regional state of Ethiopia despite of that, in this meta-analysis only four regions and one administrative city was represented in this meta-an1aysis. In addition, this could be due to differences in the socio-demographic characteristics of the study population.

The subgroup analysis of this systemic review and meta-analysis revealed that the pooled prevalence of minimum acceptable dietary intake among children aged 6–23 months is significantly different across regional state of Ethiopia. The lowest prevalence of minimum acceptable dietary intake of Ethiopian children aged 6–23 months were observed in Oromia regional state with the prevalence of 12.58% while the highest pooled prevalence was observed in Amhara regional state regions (21.76%). The possible reasons for these variations is due to the difference in study area and may other reasons like young infant and child feeding practice cultural variations and the number of studies include in the analysis in which Oromia regional state of Ethiopia were represented by only two studies. And also it might be due to the study settings of the included study; because study setting was one of the sources of heterogeneity in this meta-analysis.

Regarding study setting higher prevalence of MAD was observed in studies conducted in urban (47%) settings compared to studies conducted in rural settings. The finding from this review is supported by result from EMDHS report of 2019; in which children in urban area (14%) are more likely to be fed the MAD standard as compared to rural areas (10%) [8]. This could be due the difference in socio economic status such as educational level and income and

**Table 3. Meta regression to identify source of heterogeneity for the pooled prevalence of minimum acceptable diet among children age 6–23 months in Ethiopia, 2022.**

| Variables | Coefficient | 95% CI | p-value |
|---|---|---|---|
| Sample size | 7.03 | (-1.36, 15.4) | 0.086 |
| Years of publication | 0.05 | -0.05, 0.15 | 0.26 |

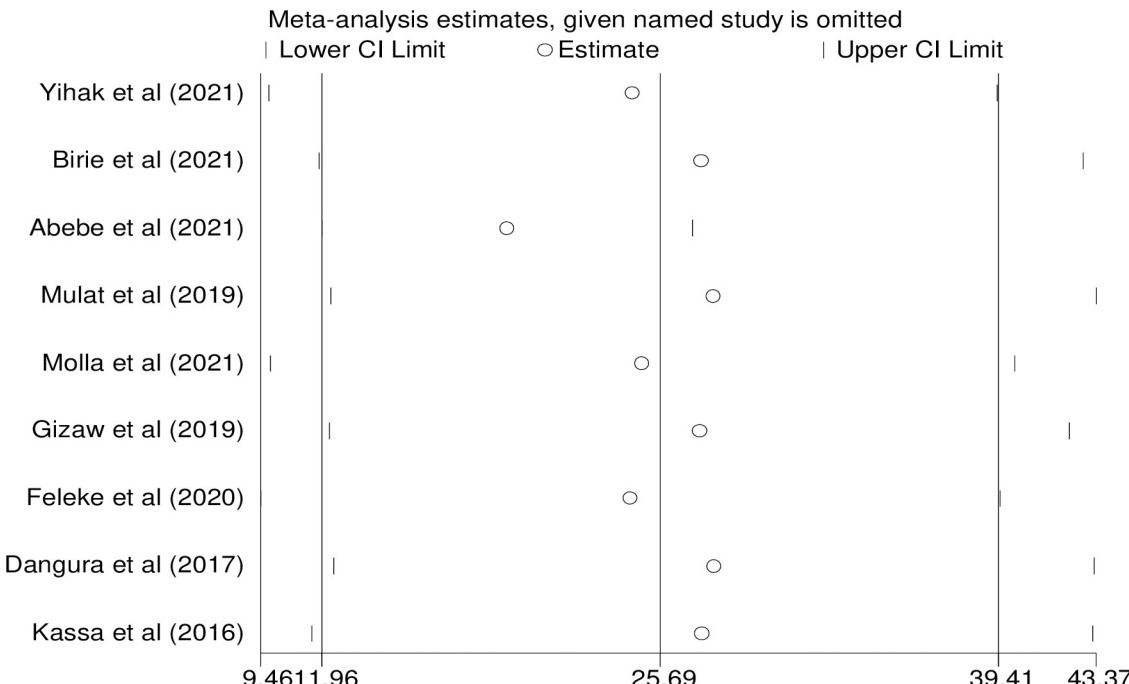

**Fig 5. Sensitivity analysis on minimum acceptable dietary intake among children aged 6 to 23 months in Ethiopia.**

also it might be related with the access of information on the benefit of health eating habit and child feeding practices.

Subgroup analysis by years of publication in this systemic review and meta-analysis also indicated that the pooled prevalence of minimum acceptable diet was relatively high on the studies published after the year 2020 was (38.4%) as compared to the studies published before 2020 (15%). This may be due to different activities and programs that have been undertaken to increase the infant and young child feeding practice. such as the recently endorsed 2019 Food and Nutrition Policy under implementation aimed to achieve optimal nutritional status throughout the life cycle by coordination nutrition-specific and nutrition sensitive interventions as their major activities [12].

## Limitation and strength of the study

The first limitations of the study was only English articles were considered for the analysis. Moreover, this meta-analysis represented only studies reported from five regions of the country. Even though we are doing sensitivity, subgroup, and meta-regression analyses to minimize the effect of heterogeneity, the degree of heterogeneity among the reviewed articles were high. Therefore, researcher and policymakers should consider effects of heterogeneity and result of filled meta-analysis during the interpretations of results.

## Conclusion and recommendation

The result of this systemic review and meta- analysis indicated that minimum acceptable diet among children aged 6–23 months was found to be low in Ethiopia; only 1 in 4children met minimum acceptable diet. Therefore, the government of Ethiopia should give greatest emphasis to meet minimum acceptable diet in order to decrease children mortality due to malnutrition as well as later consequence of in appropriate feeding practices. In relation to the global

standard national nutrition food based dietary guideline should developed by including guidance on feeding how to feed infants and young children is also recommended.

## Supporting information

**S1 Checklist. PRISMA 2020 checklist.**
(DOC)

**S1 File. Searching strategy.**
(DOCX)

## Author Contributions

**Conceptualization:** Gizachew Ambaw Kassie, Amanuel Yosef Gebrekidan, Yordanos Sisay Asgedom.

**Data curation:** Gizachew Ambaw Kassie, Amanuel Yosef Gebrekidan, Eskinder Yilma Enaro, Yordanos Sisay Asgedom.

**Formal analysis:** Gizachew Ambaw Kassie, Amanuel Yosef Gebrekidan, Yordanos Sisay Asgedom.

**Funding acquisition:** Gizachew Ambaw Kassie.

**Investigation:** Gizachew Ambaw Kassie.

**Methodology:** Gizachew Ambaw Kassie, Amanuel Yosef Gebrekidan, Eskinder Yilma Enaro, Yordanos Sisay Asgedom.

**Project administration:** Gizachew Ambaw Kassie.

**Resources:** Gizachew Ambaw Kassie, Eskinder Yilma Enaro.

**Software:** Gizachew Ambaw Kassie, Eskinder Yilma Enaro, Yordanos Sisay Asgedom.

**Supervision:** Gizachew Ambaw Kassie, Eskinder Yilma Enaro, Yordanos Sisay Asgedom.

**Validation:** Gizachew Ambaw Kassie.

**Visualization:** Gizachew Ambaw Kassie, Yordanos Sisay Asgedom.

**Writing – original draft:** Gizachew Ambaw Kassie.

**Writing – review & editing:** Gizachew Ambaw Kassie, Amanuel Yosef Gebrekidan, Eskinder Yilma Enaro, Yordanos Sisay Asgedom.

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
