## [Decision Letter · Decision Letter 0]

20 Jun 2022

PONE-D-22-14108Minimum Acceptable Dietary intake among Children Aged 6-23 Months in Ethiopia: a Systemic Review and Meta-Analysis PLOS ONE

Dear Dr. Kassie

Thank you for submitting your manuscript to PLOS ONE. After careful consideration, we feel that it has merit but does not fully meet PLOS ONE’s publication criteria as it currently stands. Therefore, we invite you to submit a revised version of the manuscript that addresses the points raised during the review process.

We look forward to receiving your revised manuscript.

Kind regards,

Yogan Pillay, Phd

Academic Editor

PLOS ONE

Journal Requirements:

2. Please ensure that you refer to Figure 4 in your text as, if accepted, production will need this reference to link the reader to the figure.

Additional Editor Comments:

As above

Reviewers' comments:

Reviewer's Responses to Questions

**Comments to the Author**

1. Is the manuscript technically sound, and do the data support the conclusions?

Reviewer #1: Yes

2. Has the statistical analysis been performed appropriately and rigorously? 

Reviewer #1: Yes

3. Have the authors made all data underlying the findings in their manuscript fully available?

Reviewer #1: Yes

4. Is the manuscript presented in an intelligible fashion and written in standard English?

Reviewer #1: Yes

5. Review Comments to the Author

Reviewer #1: This is a noteworthy topic and methodologically sound manucript. There are a few minor corrections and amendments needed. I have uploaded my comments in a separate document which indicates the needed corrections or ammendments by line in the manuscript.

6. PLOS authors have the option to publish the peer review history of their article (what does this mean?). If published, this will include your full peer review and any attached files.

Reviewer #1: No

---

## [Author Response · Author response to Decision Letter 0]

8 Jul 2022

July 8, 2022

Yogan Pillay, PhD

Academic Editor

PLOS ONE

RE: PONE-D-22-14108: Minimum Acceptable Dietary among Children Aged 6-23 Months in Ethiopia: a Systemic Review and Meta-analysis 

Dear Dr. Yogan Pillay, 

We thank you for considering our manuscript and for arranging for it to be reviewed by reviewer. We have tried to address your comments and the comments / suggestions from the reviewer. We have also incorporated changes to reflect most of the suggestion provided by you and the reviewer. We have highlighted the changes within the manuscript please find for your kind consideration the following: In the Response to Reviewers, we copy each of the comments / suggestions and provide the RESPONSE hereunder. We also provide a marked-up copy of the manuscript that highlights changes made to the original version and this is uploaded as a separate file labelled “Revised Manuscript with Track Changes”. Finally, we provide an unmarked version of the revised manuscript without tracked changes and this is uploaded as a separate file labelled 'Manuscript'. 

We have been carefully through the peer review and have revised our paper accordingly. We feel that the paper is much improved as a result of this peer review process, and thank you for taking it to this stage. While hoping that these changes would meet with your favorable consideration, we hold ourselves at your entire disposition for any further information or other changes you might require. The authors welcome further constructive comments.

Sincerely yours! 

Gizachew Ambaw Kassie on behalf of the co-authors

POINT BY POINT RESPONSE TO THE EDITOR AND REVIEWER COMMENTS

RESPONSE TO ACADEMIC EDITOR:

RESPONSE: We have checked and have ensured that the manuscript met the journal’s requirements. 

2. Please ensure that you refer to Figure 4 in your text as, if accepted, production will need this reference to link the reader to the figure

RESPONSE: Thank you for pointing out this point. We accept and revision has been made in the main document. 

Please include captions for your supporting information files at the end of your manuscript, and update any in-text citations to match accordingly 

RESPONSE: Thank you; we have included captions as supporting information at the end of our manuscript 

POINT BY POINT RESPONSE TO REVIEWER

Well written and methodological sound manuscript. A few minor corrections:

RESPONSE: We thank the reviewer for kind work and constructive comments. We have revised our manuscripts based on the reviewer’s comments as below. 

Line 45: Better to say 1 in 4 children rather than one fourth of children

RESPONSE: Thank you, we have replaced the term as suggested

Line 81: Sentence needs restructuring to read as follows:- ‘This indicates that feeding of the minimum acceptable diet is a major problem in Ethiopia. You cannot say global problem when you have only presented data for Ethiopia

RESPONSE: Agree, we have modified accordingly 

Line 170: Capital letters for Egger’s and Begg

RESPONSE: Thanks for this comment, corrected 

Line 191: Define cases and Prevalence in your table. What cases and prevalence of what?

RESPONSE: Thanks for the this suggestion, we have defined cases in the main document as “children who met Minimum Acceptable Diet (MAD)” and prevalence as “Prevalence of MAD” 

Line 235: restructure sentence to read ‘national Demographic and Health surveys in China, ….

RESPONSE: Thank you for your suggestion, the sentence have restructured as “The result of this meta-analysis was consistent with the national Demographic and Health surveys in China”

Line 237: remove ‘finding from’

RESPONSE: Thank you, noted and deleted 

Line 238: ‘in low- and middle-income countries…

RESPONSE: Comment accepted and well treated 

Line 234: heterogeneity without capital H

RESPONSE: Thank you, accepted 

Line 256: Correction: higher prevalence of MAD was observed in studies conducted in urban settings compared to studies conducted in rural settings. 

RESPONSE: Thank you, we have replaced the sentence as indicated

Line 265: restructure sentence to read ‘more likely to be fed the MAD standard…’

RESPONSE: Comment accepted and revised as recommended 

Line 268: delete after the end of the MDGs

RESPONSE: We thank you again, we have edited accordingly 

Line 268-271: This sentence does not add any value to the discussion of IYCF or MAD. Or restructure to say how the new 2019 Nutrition programme is addressing IYCF or MAD.

RESPONSE: Thank you for insightful comment. The sentence putted prior to this information was not related with this discussion that is why it was confused. Therefore, the indicated sentence is moved to the preceding paragraph and I hope the information from “line 268-271” add a value as a justification why MAD is higher in the studies published after the year 2020 which might be due to the implementation of new 2019 food and nutrition policy in Ethiopia aimed to achieve optimal nutritional status throughout the life cycle. We also made some restructuring of the sentence. 

Line 277: 1 in 4 children rather than ‘one fourth of children…’

RESPONSE: Thank you, we have replaced the term as suggested

---

## [Editor Report · Decision Letter 1]

2 Aug 2022

PONE-D-22-14108R1Minimum Acceptable Dietary intake among Children Aged 6-23 Months in Ethiopia: a Systemic Review and Meta-AnalysisPLOS ONE

Dear Dr.Kassie

Thank you for submitting your manuscript to PLOS ONE. After careful consideration, we feel that it has merit but does not fully meet PLOS ONE’s publication criteria as it currently stands. Therefore, we invite you to submit a revised version of the manuscript that addresses the points raised during the review process.

We look forward to receiving your revised manuscript.

Kind regards,

Yogan Pillay, Phd

Academic Editor

PLOS ONE

---

## [Author Response · Author response to Decision Letter 1]

9 Aug 2022

POINT BY POINT RESPONSE TO REVIEWER

Well written and methodological sound manuscript. A few minor corrections:

RESPONSE: We thank the reviewer for kind work and constructive comments. We have revised our manuscripts based on the reviewer’s comments as below. 

Line 45: Better to say 1 in 4 children rather than one fourth of children

RESPONSE: Thank you, we have replaced the term as suggested

Line 81: Sentence needs restructuring to read as follows:- ‘This indicates that feeding of the minimum acceptable diet is a major problem in Ethiopia. You cannot say global problem when you have only presented data for Ethiopia

RESPONSE: Agree, we have modified accordingly 

Line 170: Capital letters for Egger’s and Begg

RESPONSE: Thanks for this comment, corrected 

Line 191: Define cases and Prevalence in your table. What cases and prevalence of what?

RESPONSE: Thanks for the this suggestion, we have defined cases in the main document as “children who met Minimum Acceptable Diet (MAD)” and prevalence as “Prevalence of MAD” 

Line 235: restructure sentence to read ‘national Demographic and Health surveys in China, ….

RESPONSE: Thank you for your suggestion, the sentence have restructured as “The result of this meta-analysis was consistent with the national Demographic and Health surveys in China”

Line 237: remove ‘finding from’

RESPONSE: Thank you, noted and deleted 

Line 238: ‘in low- and middle-income countries…

RESPONSE: Comment accepted and well treated 

Line 234: heterogeneity without capital H

RESPONSE: Thank you, accepted 

Line 256: Correction: higher prevalence of MAD was observed in studies conducted in urban settings compared to studies conducted in rural settings. 

RESPONSE: Thank you, we have replaced the sentence as indicated

Line 265: restructure sentence to read ‘more likely to be fed the MAD standard…’

RESPONSE: Comment accepted and revised as recommended 

Line 268: delete after the end of the MDGs

RESPONSE: We thank you again, we have edited accordingly 

Line 268-271: This sentence does not add any value to the discussion of IYCF or MAD. Or restructure to say how the new 2019 Nutrition programme is addressing IYCF or MAD.

RESPONSE: Thank you for insightful comment. The sentence putted prior to this information was not related with this discussion that is why it was confused. Therefore, the indicated sentence is moved to the preceding paragraph and I hope the information from “line 268-271” add a value as a justification why MAD is higher in the studies published after the year 2020 which might be due to the implementation of new 2019 food and nutrition policy in Ethiopia aimed to achieve optimal nutritional status throughout the life cycle. We also made some restructuring of the sentence. 

Line 277: 1 in 4 children rather than ‘one fourth of children…’

RESPONSE: Thank you, we have replaced the term as suggested

---

## [Decision Letter · Decision Letter 2]

23 Nov 2022

PONE-D-22-14108R2Minimum Acceptable Dietary intake among Children Aged 6-23 Months in Ethiopia: a Systemic Review and Meta-AnalysisPLOS ONE

Dear Dr. Gizachew Ambaw Kassie,

Thank you for submitting your manuscript to PLOS ONE. After careful consideration, we feel that it has merit but does not fully meet PLOS ONE’s publication criteria as it currently stands. Therefore, we invite you to submit a revised version of the manuscript that addresses the points raised during the review process.

We look forward to receiving your revised manuscript.

Kind regards,

Yasin Sahin

Academic Editor

PLOS ONE

Reviewers' comments:

Reviewer's Responses to Questions

**Comments to the Author**

1. If the authors have adequately addressed your comments raised in a previous round of review and you feel that this manuscript is now acceptable for publication, you may indicate that here to bypass the “Comments to the Author” section, enter your conflict of interest statement in the “Confidential to Editor” section, and submit your "Accept" recommendation.

Reviewer #2: All comments have been addressed

Reviewer #3: (No Response)

2. Is the manuscript technically sound, and do the data support the conclusions?

Reviewer #2: No

Reviewer #3: Partly

3. Has the statistical analysis been performed appropriately and rigorously? 

Reviewer #2: No

Reviewer #3: No

4. Have the authors made all data underlying the findings in their manuscript fully available?

Reviewer #2: (No Response)

Reviewer #3: Yes

5. Is the manuscript presented in an intelligible fashion and written in standard English?

Reviewer #2: No

Reviewer #3: No

6. Review Comments to the Author

Reviewer #2: The rationale for the present meta-analysis is rather weak and unclear. The pooled prevalence estimate is not useful or informative given the very substantial heterogeneity (I2 > 99%) among the different studies and contexts being pooled.

Specific comments:

1. This is a "systematic review" and not a "Systemic Review".

2. Please change "Even though, many studies" to "Even though many studies".

3. Please change "The Global burden" to "The global burden".

4. As per PRISMA guidelines, please specify in the methods section if the review protocol was prospectively registered. Indicate if a review protocol exists, if and where it can be accessed (e.g., Web address).

5. The reliability and reproducibility of a systematic review cannot be overemphasized. Please also provide the full electronic search strategy used to identify studies, including all search terms and limits for at least one database. The rest can be provided in the supplementary material.

6. Please change "gray literature" to "grey literature".

7. Please change "Homogeneity across studies" to "Heterogeneity across the studies".

8. "Full title articles excluded with reason (n=21)" - please state the exact reasons for exclusion.

9. It was unclear if the samples were drawn from population or clinical samples. This should be stated as clinical samples would be quite different in characteristics.

10. The findings of the result were not well discussed. What minimum dietary intake value do the authors consider to be acceptable, considering the characteristics of the country, the population and the environment? What specific recommendations can be made to address these gaps?

11. The discussion of study limitations was grossly inadequate. How can we account for the very significant heterogeneity in the meta-analysis? Equally if not more important than the point estimate, the distribution of estimates (i.e., how much they are dispersed around the average pooled estimate) is a key aspect in meta-analyses of prevalence. Are the studies truly comparable?

12. "We would like to thank all of the authors of studies participated in this systematic review and meta-analysis" - this does not make sense. Suggest to omit.

Reviewer #3: General comments

1. The work lacks nobility. Since there is EDHS 2019 that reported a prevalence of 11% minimum acceptable diet, it is not possible to generate a pooled prevalence that is more valid and reliable than the EDHS 2019 finding. How can your finding be preferred over EDHS 2019 finding. Your pooled prevalence is very far to represent the national prevalence as the included studies are few and obtained from only four regions out of a total of 11 regions in the country.

2. I suggest a systematic review only. You should not pool a prevalence (never conduct a meta-analysis).

3. There is a serious English language editorial problem

Specific comments

Abstract

Introduction:

Line 27: remove “an adequate” as “minimum acceptable diet” is by itself enough to convey the intended message.

Line 28 and second sentence: It says “Feeding the minimum acceptable…………” Correct as “Not feeding at least the minimum acceptable diet…….’’

Line 30: Your objective was to produce a pooled prevalence as a result of a rationale of inconsistence studies in Ethiopia. The problem you actually observed from your result is a serious consistence across studies as evidenced by your heterogeneity test. So, I recommend you end up with systematic review only rather than pooling a very heterogeneous results, which may be due to small number of studies.

Method:

Line 35: “Data was” correct as “Data were”

Results:

What was the source of heterogeneity since you mentioned in the method section above that you did a subgroup analysis to identify the source of heterogeneity?

Key words (incorrect), Keyword (correct): Separate the keywords by a semicolon not by a comma.

Introduction

The introduction content need to be re-structured so as to create a logical follow of the introduction information.

As there was 2019 EDHS’s report of 11% minimum acceptable diet proportion, how strong your finding will be in attracting the governments attention as compared to the EDHS 2019 finding? Due to this and other reasons, I suggest you simply conduct systematic review only.

Line108 and 109: Published and unpublished reports: There should be consistency. Which one you really did in your review? Only published (in the abstract) or both Published and Unpublished (introduction).

Line 116: Which searching terms were combined by "and" which were combined by "or"?

Line 127: Exclusion criteria – not published in English was already mention in the inclusion criteria.

Line 137: What is that standardized data abstraction sheet you used?

Results

The nine studies are all from three regions and one city administration. So, meta-analysis is not important.

Line 191: Remove meta-analysis in the bracket.

Discussion

Line 232: Please cite for the global recommendations at the end of “This Indicates that most children aged 6–23 months in Ethiopia were not taking the minimum acceptable diet according to global recommendations”.

Line 234-235: You compared your finding with other Countries’ DHS report. So, the Ethiopian Demographic and Health Survey finding is more comparable with those DHS reports than your pooled prevalence. So, I recommend only systematic review.

Line 236: You again compared your finding with the EDHS 2019 report and yours was higher. The explanation you gave for the difference was your study was not representative of all regions as EDHS was. So, this shows your study was not important at all. Because the EDHS finding is valid than yours.

Limitation and strength

Line 271: Why you only included English language articles?

Line 272: Since we already the EDHS 2019 report that has not the limitations encountered by your study, EDHS is more important than yours.

Conclusion and recommendation

“The result of this systemic review and meta- analysis indicated that minimum acceptable diet among children aged 6-23 months was found to be low in Ethiopia; only 1 in 4children met the minimum acceptable diet.”

Your study finding (25%) is better than EDHS finding (11%). So, your recommendation is not acceptable since it has serious limitations.

References

References are not edited and formatted

7. PLOS authors have the option to publish the peer review history of their article (what does this mean?). If published, this will include your full peer review and any attached files.

Reviewer #2: No

Reviewer #3: No

---

## [Author Response · Author response to Decision Letter 2]

30 May 2023

Reviewer #2: 

General comment: The rationale for the present meta-analysis is rather weak and unclear. The pooled prevalence estimate is not useful or informative given the very substantial heterogeneity (I2 > 99%) among the different studies and contexts being pooled.

Response: Dear reviewer, we would like to express our gratitude for the tie you have taken to insightful and constructive comments on our manuscript, and we appreciate the effort you have put in to review our paper. Yes, of course high heterogeneity can affect our result. However to address this significant heterogeneity; we have estimated the pooled prevalence by random effect model. As well as we performed subgroup analysis and meta-regression to identify the possible cause of the heterogeneity. In addition, we acknowledge the limitations and uncertainties associated with meta-analyses and to interpret the findings cautiously in light of the available evidence.

Specific comments:

Comment 1: This is a "systematic review" and not a "Systemic Review".

Response: Thank you for pointing out it. We have corrected it in the revised version

Comment 2: Please change "Even though, many study" to "Even though many studies".

Response: Thank you! Comment accepted and corrected as recommended.

Comment 3: Please change "The Global burden" to "The global burden".

Response: Thank you, comment accepted and well treated accordingly.

Comment 4: As per PRISMA guidelines, please specify in the methods section if the review protocol was prospectively registered. Indicate if a review protocol exists, if and where it can be accessed (e.g., Web address).

Response: Thank you again for suggestion and comment. We have stated as “the protocol is was not registered 

Comment 5: The reliability and reproducibility of a systematic review cannot be overemphasized. Please also provide the full electronic search strategy used to identify studies, including all search terms and limits for at least one database. The rest can be provided in the supplementary material.

Response: Thank you again for suggestion and comment. We have included the searching terms for PubMed in the main manuscripts in addition. We have also attached the searching strategy as supplementary files.

Comment 6: Please change "gray literature" to "grey literature".

Response: Comment accepted and corrected as “grey”

Comment 7: Please change "Homogeneity across studies" to "Heterogeneity across the studies".

Response: Thank you for pointing out it. Corrected in the revised manuscript

Comment 8: "Full title articles excluded with reason (n=21)" - please state the exact reasons for exclusion.

Response: Thank you, comment accepted and well treated accordingly. 

Comment 10: The findings of the result were not well discussed. What minimum dietary intake value do the authors consider to be acceptable, considering the characteristics of the country, the population and the environment? What specific recommendations can be made to address these gaps?

Response: Thank you! We have added limitation regarding heterogeneity 

Comment 11: The discussion of study limitations was grossly inadequate. How can we account for the very significant heterogeneity in the meta-analysis? Equally if not more important than the point estimate, the distribution of estimates (i.e., how much they are dispersed around the average pooled estimate) is a key aspect in meta-analyses of prevalence. Are the studies truly comparable?

Response: Thank you, we have made extensive revision 

Comment 12: "We would like to thank all of the authors of studies participated in this systematic review and meta-analysis" - this does not make sense. Suggest to omit.

Response: Thank you pointing out this error. We have omit it based on the comment 

Reviewer #3: 

General comments

Comment 1: The work lacks nobility. Since there is EDHS 2019 that reported a prevalence of 11% minimum acceptable diet, it is not possible to generate a pooled prevalence that is more valid and reliable than the EDHS 2019 finding. How can your finding be preferred over EDHS 2019 findings? Your pooled prevalence is very far to represent the national prevalence as the included studies are few and obtained from only four regions out of a total of 11 regions in the country.

Response: Thank you reviewer for devoting your time to review our manuscript and your comments and suggestions. We have corrected it in the revised version of the manuscript. We have stated this issue as a limitation. 

Comment 2: I suggest a systematic review only. You should not pool prevalence (never conduct a meta-analysis).

Response: Thank you very much for you suggestions. Even though, some limitation of this meta-analysis, we believe that our finding has a greatest implication for policy maker because meta-analysis increase the power of the study 

Comment 3: there is a serious English language editorial problem

Response: Thank you, we have checked grammatically errors, writing style and typos; and made extensive revisions in the revised manuscript. 

Specific comments

Abstract

Introduction:

Comment: Line 27: remove “an adequate” as “minimum acceptable diet” is by itself enough to convey the intended message.

Response: Thank you for pointing out this comment. We have revised it accordingly in the revised manuscript

Comment: Line 28 and second sentence: It says “Feeding the minimum acceptable…………” Correct as “Not feeding at least the minimum acceptable diet…….’’

Response: Thank you for pointing out it. Corrected based on the comment

Comment: Line 30: Your objective was to produce a pooled prevalence as a result of a rationale of inconsistence studies in Ethiopia. The problem you actually observed from your result is a serious consistence across studies as evidenced by your heterogeneity test. So, I recommend you end up with systematic review only rather than pooling a very heterogeneous results which may be due to small number of studies.

Response: Thank you for bringing this issue. To address these limitations, we carefully assess the sources of heterogeneity, conduct subgroup analyses, use appropriate statistical methods, and report the results transparently and comprehensively. We have acknowledge the limitations and uncertainties associated with meta-analyses and to interpret the findings cautiously in light of the available evidence

Method:

Comment: Line 35: “Data was” correct as “Data were”

Response: Thank you pointing out the error. We have corrected it 

Results:

Comment: What was the source of heterogeneity since you mentioned in the method section above that you did a subgroup analysis to identify the source of heterogeneity?

Response: Thank you for your question. The observed high heterogeneity might be due to the difference in the study area, study period, residence and other unspecified variations.

Comment: Key words (incorrect), Keyword (correct): Separate the keywords by a semicolon not by a comma.

Response: thank you we have revised it 

Comment: As there was 2019 EDHS’s report of 11% minimum acceptable diet proportion, how strong your finding will be in attracting the governments attention as compared to the EDHS 2019 finding? Due to this and other reasons, I suggest you simply conduct systematic review only.

Response: Dear reviewer, we would like to thank you for your suggestions. However, despite searching different databases, we were unable to find systematic reviews or meta-analyses on the minimum acceptable diet. Therefore, the papers reviewed were discussed with those of the national studies

Comment: Line108 and 109: Published and unpublished reports: There should be consistency. Which one you really did in your review? Only published (in the abstract) or both Published and Unpublished (introduction).

Response: we apologize for the inconvenience. We have corrected it based on your comment 

Comment: Line 116: Which searching terms were combined by "and" which were combined by "or"?

Response: Comment accepted, we have revised it in the revised draft

Comment: Line 127: Exclusion criteria – not published in English was already mention in the inclusion criteria.

Response: Thank you for the comment. We included articles published only in English language for the sake of clarity, understandability, and simplicity of interpretations. In addition, we have stated as a limitation 

Comment: Line 137: What is that standardized data abstraction sheet you used?

Response: We used excel spreadsheet to extract the data

Results

Comment: The nine studies are all from three regions and one city administration. So, meta-analysis is not important.

Response: Thank you for your suggestion. 

Comment: Line 191: Remove meta-analysis in the bracket.

Response: Thank you, we have removed it 

Discussion

Comment: Line 232: Please cite for the global recommendations at the end of “This Indicates that most children aged 6–23 months in Ethiopia were not taking the minimum acceptable diet according to global recommendations”

Response: than you we have cited appropriately. 

Comment: Line 234-235: You compared your finding with other Countries’ DHS report. So, the Ethiopian Demographic and Health Survey finding is more comparable with those DHS reports than your pooled prevalence. So, I recommend only systematic review.

Response: Despite searching different databases, we were unable to find systematic reviews or meta-analyses on the minimum acceptable diet. Therefore, the papers reviewed were discussed with those of the national studies

Comment: Line 236: You again compared your finding with the EDHS 2019 report and yours was higher. The explanation you gave for the difference was your study was not representative of all regions as EDHS was. So, this shows your study was not important at all. Because, the EDHS finding is valid than yours

Response: Thank you very much for your interesting comments and suggestions. One important thing for a research is creating a question for other researchers in order to conduct further research on that specific problem. To the best of my knowledge, it is the first meta-analysis and the finding too. Hence, we give a justification about its discrepancy. In addition, we have mention the problem in the limitation.

Limitation and strength

Comment: Line 271: Why you only included English language articles?

Response: Thank you for the comment. We included articles published only in English language for the sake of clarity, understandability, and simplicity of interpretations.

Comment: Line 272: Since us already the EDHS 2019 report that has not the limitations encountered by your study, EDHS is more important than yours.

Response: dear reviewer thank you for your comment. As we know one of the key characteristics of research is a single study cannot stand alone. We need to have different research finding with different approaches in order to implement appropriate intervention mechanisms 

Conclusion and recommendation

Comment: “The result of this systemic review and meta- analysis indicated that minimum acceptable diet among children aged 6-23 months was found to be low in Ethiopia; only 1 in 4children met the minimum acceptable diet.” Your study finding (25%) is better than EDHS finding (11%). So, your recommendation is not acceptable since it has serious limitations.

Response: Thank you. We compare our finding based on the WHO and other guideline recommendation. The World Health Organization (WHO) recommends ensuring a minimum acceptable diet (MAD) for all children, particularly in the early years, to promote healthy growth and development

References

Comment: References are not edited and formatted

Response: Thank you, we have edited and we have corrected the formats based on your comment

---

## [Editor Report · Decision Letter 3]

2 Jun 2023

Minimum Acceptable Dietary intake among Children Aged 6-23 Months in Ethiopia: a Systematic Review and Meta-Analysis

PONE-D-22-14108R3

Dear Dr. Gizachew Ambaw Kassie,

We’re pleased to inform you that your manuscript has been judged scientifically suitable for publication and will be formally accepted for publication once it meets all outstanding technical requirements.

Kind regards,

Yasin Sahin

Academic Editor

PLOS ONE

Additional Editor Comments (optional):

Thanks to the authors for the study. I think that it will contribute to the literature.
---

## [Editor Report · Acceptance letter]

7 Jun 2023

PONE-D-22-14108R3 

Minimum acceptable dietary intake among children aged 6-23 months in Ethiopia: a Systematic review and meta-analysis 

Dear Dr. Kassie:

I'm pleased to inform you that your manuscript has been deemed suitable for publication in PLOS ONE. Congratulations! Your manuscript is now with our production department. 

Kind regards, 

on behalf of

Dr. Yasin Sahin 

Academic Editor

PLOS ONE